# Refractive error and its associated factors among pregnant women attending antenatal care unit at the University of Gondar Comprehensive Specialized Hospital, Northwest Ethiopia

**Mengistie Diress[ID][1]\*, Yigizie Yeshaw[1,2], Minychil Bantihun[3], Baye Dagnew[1], Adugnaw Ambelu[1], Mohammed Abdu Seid[4], Yonas Akalu[1]**

**1** Department of Human Physiology, School of Medicine, University of Gondar, Gondar, Ethiopia,
**2** Department of Epidemiology and Biostatistics, Institute of Public Health, University of Gondar, Gondar, Ethiopia, **3** Department of Optometry, School of Medicine, University of Gondar, Gondar, Ethiopia,
**4** Department of Human Physiology, School of Medicine, Debre Tabor University, Debre Tabor, Ethiopia

\* mengistiediress@gmail.com

## Abstract

### Background

Refractive error is one of the commonly encountered problems during pregnancy and being the cause of deleterious effects on health. Despite its impacts, there is no evidence on the magnitude and associated factors of refractive error among pregnant women in Ethiopia. This study aimed to determine the prevalence of refractive error and its associated factors among pregnant women attending antenatal care unit at the University of Gondar Comprehensive Specialized Hospital, Northwest Ethiopia, 2020.

### Methods

An institution-based cross-sectional study was employed. An ocular examination was performed using Retinoscope and Snellen's illiterate "E" chart. The required data were collected using an interviewer-administered questionnaire which comprised socio-demographic, clinical and pregnancy-related variables. EpiData 3.02 and STATA 14 were used for data entry and analysis respectively. Both bivariable and multivariable binary logistic regression analyses were executed to identify factors associated with refractive error. Variables with a p-value $\leq$ 0.05 in the multivariable logistic regression analysis were declared as significantly associated factors with refractive error.

### Results

A total of 401 pregnant women with a median age of 27 (IQR = 24–31) years participated in this study. The overall prevalence of refractive error among the study participants was 35.66% (95% CI: 30.95–40.37). Of the total study participants, ninety-two (22.90%) of them were myopic, forty-five (11.22%) were hyperopic and the rest were antimetropic. Increased

**Data Availability Statement:** All relevant data are within the paper and its Supporting Informationw files.

**Funding:** The author(s) received no specific funding for this work.

**Competing interests:** The authors have declared that no competing interests exist.

**Abbreviations:** ANC, Ante-Natal Care; D, Diopters; GDM, Gestational Diabetes Mellitus; PIH, Pregnancy Induced Hypertension; SE, Spherical Equivalent; UoG, University of Gondar; UoGCSH, University of Gondar Comprehensive Specialized Hospital; WHO, World Health Organization.

maternal age (AOR = 1.31; 95% CI: 1.16–1.48)), increased parity (AOR = 3.17, 95% CI: 1.92–5.25), increased gestational age (AOR = 1.15, 95% CI: 1.08–1.22), and regular use of computers/ watching television (AOR = 6.19, 95% CI: 2.46–15.59) were significantly associated with refractive error.

## Conclusion

The prevalence of refractive error among pregnant women was high where myopia was the most common variety. Advanced maternal age, increased gestational age, increased parity and regular use of computer or watching television were significantly associated with refractive error among pregnant women. Therefore, apart from providing other maternal health services, routine screening and evaluation of pregnant women for refractive error during antenatal care visit is recommended to avoid its negative impacts.

## Introduction

Refractive error (RE) is the leading cause of visual impairment worldwide [1–3]. Globally, 333 million people are visually impaired. Of these, 153 million cases are attributed to uncorrected refractive error [2,3]. The prevalence of RE is about 46% in Sub-Saharan Africa [3,4]. According to the 2006 National Survey Report of Ethiopia, RE is the second cause of visual impairment (33.40%) following cataract (42.3%) and the third cause of blindness (7.80%) [5].

Studies in the world consistently showed that females have a relatively higher risk of developing RE than males [6,7]. The latest study in Saudi Arabia estimated that the prevalence of RE among women is 25.36% [8]. RE is also a common problem among pregnant women in developing countries [9,10]. A study in India revealed that 65% of pregnant women have myopia [11]. The finding of another study in Nigeria shows that myopia is the most prevalent type of RE (57%) followed by hyperopia (33%) [10]. The prevalence of RE in South India is estimated to be 77.50% [12].

Uncorrected RE among pregnant women is associated with both immediate and long-term consequences. This problem leads to disabilities, unable to travel safely through the environment, loss of educational and job opportunities, and reduced productivity among the women. It can also spoil the quality of life of the women and may give rise to retinal detachment if not treated early [13–17].

Evidences from different regions of the world revealed that RE is associated with maternal age [18,19], residence [20], level of education [21,22], gestational age [10,23,24], mental stress [25], sleep disturbance [26], family history of RE [27], contraceptive use [28], iron deficiency anemia [29], gestational diabetes mellitus (GDM) [30,31], pregnancy induced hypertension (PIH) [32,33], medication history [34], and prolonged use of digital devices [1,35,36].

Though various literatures revealed the existence of numerous pregnancy-associated visual problems, little is known about RE in Africa particularly in Ethiopia. Therefore, this study aimed to determine the prevalence and associated factors of RE among pregnant women attending antenatal care (ANC) unit at the University of Gondar Comprehensive Specialized Hospital (UoGCSH). Conducting this research will have a crucial role in filling the above gaps, and to design and implement strategies for the prevention and management of the problem in the study setting.

## Methods and materials

### Ethical approval and consent to participate

Ethical clearance with a reference number of 1828/2012 was obtained from the Institutional Review Board (IRB) of the UoG and letter of cooperation was gotten from the UoGCSH. Written informed consent was taken from each study participant. Privacy and confidentiality of information were kept properly and names were not recorded. Those study participants who had severe RE at the time of data collection were referred to the Department of Ophthalmology for further diagnosis and management.

### Study setting, design, and population

An institution-based cross-sectional study design was carried out at the UoGCSH, ANC unit from 01 February 2020, to 30 March 2020. The University of Gondar (UoG) is located 727 km far from the capital city, Addis Ababa in the Northwest direction. ANC unit of the UoGCSH is one of the outpatient departments established earlier and currently provides services for about 22,824 pregnant women per year. Pregnant women visiting the ANC service are regularly evaluated and the findings were documented in their charts. All pregnant women who had ANC service at the UoGCSH were included in the study whereas, pregnant women with known diabetes mellitus and hypertension before being pregnant, those with congenital eye problems, refractive problems, glaucoma, cataract, and eye trauma during the study period were excluded from the study.

### Sample size determination and sampling procedure

The required sample size for the study was estimated using a single population proportion formula, taking an expected estimate of the prevalence of 50% (no previous study was conducted in Ethiopia), 95% confidence interval, and a 5% margin of error. Accordingly, the sample size for this study became 384.

$$N = \frac{(Z\alpha/2)^2 * p(1-p)}{d^2} = \frac{(1.96)^2 * 0.5(1-0.5)}{(0.5)^2} = 384$$

Where; N = sample size, p = expected estimates of prevalence value (50%), d = margin of sampling error tolerated- 5% (0.05), $Z\alpha/2$ = critical value at 95% confidence interval of certainty (1.96).

After adding a 5% non-response rate, the final sample size became 403. Then, a systematic random sampling technique was used to recruit the study participants. The first participant was recruited by lottery method and the remaining participants were recruited by adding 9 (interval size) which was determined by dividing the total number of pregnant women who attended ANC unit during the study period (3804) to the required sample size (403).

### Study variables

The dependent variable was refractive error (yes or no). Participants were considered to have a refractive error, if they were either myopic (the spherical equivalent ≤ –0.50 diopters (D)) or hyperopic (the spherical equivalent ≥ +0.50D) or both problems in either or both eyes. If the spherical equivalent (SE) was in between -0.50D and +0.50D in either or both eyes, they were considered as normal [37,38].

The independent variables for this study were age, residence, income, occupation, educational level, parity, gestational age, GDM, preeclampsia, and eclampsia, Catha edulis (khat),

cigarette smoking, coffee drink, alcohol intake, anemia, medication history, stress, sleep disturbance, family history of spectacle use, and regular use of computers or watching TV.

## Operational definitions

**Refractive error.** A spherical equivalent (SE) of less than –0.50 and/or greater than +0.50D in either eye or both [37,38].

**Myopia.** A spherical equivalent (SE) of less than –0.50D in either eye or both, that can be corrected by concave lens [38,39].

**Hyperopia.** A spherical equivalent (SE) of greater than +0.50D in either eye or both, that can be corrected by convex lens [38].

**Antimetropia.** A situation in which one eye is hyperopic, while the other is myopic [40].

**Anemia.** Hemoglobin level of less than 12.00 g/dl or hematocrit level of less than 36.00% [41].

**Sleep disturbance.** Sleeping time of $\leq$ 5 hours/day or sleeping time of $\geq$ 9 hours/day [42,43].

**Substance use.** The use of at least one of the substances (alcohol, khat/Catha edulis, cigarettes, and coffee) in an individual's lifetime.

**Current substance user.** A person who consumed substances (alcohol, khat, cigarettes, and coffee) at least once within the last 30 days.

**Substance ever uses.** Use of any of the substances (alcohol, khat, cigarettes, and coffee) at least once in an individual's lifetime.

**Regular use of computers or television.** Reading or watching of computers or television at least once a day for not less than 2 hours [35].

**Medication history.** Taking of anti-rheumatic, anti-psychiatric & anti-thrombotic drugs in the last one month.

## Data collection tools and procedure

Data were collected using a validated interviewer-administered questionnaire comprised of socio-demographic information (age, occupation, educational level, and residence), pregnancy-related characteristics (gestational age, GDM, and parity), and history of medical conditions (anemia, stress, sleep disturbance, and medication history). Visual acuity test (both unaided and with current correction,) was measured by Snellen's illiterate "E" chart in a well-illuminated room, from the women positioned at a 6-meter distance from the chart. Non-cycloplegic refraction was performed using trial lenses, trial frame, and retinoscopy in a semi-dark examination room. Adult sized blood pressure cuff was used to measure systolic and diastolic blood pressure. The data were collected by two BSc Midwives and two Optometrists.

## Data quality management

To assure the data quality, high emphasis was given in designing data collection tools. A pretest was done at Polly Health Center at Gondar town before the actual data collection. The questionnaire was prepared in English and then translated to Amharic (local language) by language expert and retranslated to English by another language expert to check its consistency. Training was also given for the data collectors and supervisor regarding the purpose of the study, ethical issues, and interview and measurement techniques. Strict supervision was undertaken throughout the data collection. The supervisor and principal investigator daily checked the questionnaire for accuracy and completeness.

## Data processing and analysis procedure

The collected data were checked for completeness, then entered into EpiData version 3.02 and analyzed using STATA 14. Descriptive measures such as median, frequency, and interquartile range (IQR) were calculated. Both bivariable and multivariable binary logistic regression analyses were performed. Those variables with a p-value of < 0.25 in the bivariable analysis were entered into multivariable logistic regression. We also determined the Adjusted Odds Ratio (AOR) with its 95% confidence interval for those variables included in the multivariable logistic regression model. In the final model, variables with a p-value ≤ 0.05 were declared as significantly associated with RE.

## Ethical approval and consent to participate

Ethical clearance with a reference number of 1828/2012 was obtained from the Institutional Review Board (IRB) of the UoG and letter of cooperation was gotten from the UoGCSH. Written informed consent was taken from each study participant. Privacy and confidentiality of information were kept properly and names were not recorded. Those study participants who had severe RE at the time of data collection were referred to the Department of Ophthalmology for further diagnosis and management.

# Results

## Socio-demographic characteristics of the study participants

A total of 401 pregnant women participated in this study with a response rate of 99.50%. The age range of participants was from 17 to 43 years with a median age of 27 (IQR = 24–31) years. Majority of the respondents were Christianity followers (79.80%) and urban dwellers (83.54%). Regarding educational status, 151 (37.66%) had an education level of college/university (Table 1).

## Clinical, substance use and pregnancy-related characteristics of the study participants

The gestational age of pregnant women ranged from 4 to 40 weeks. Nineteen (4.74%) of the study participants had GDM and about 7% had a history of pregnancy-induced hypertension. Majority of the study participants (63.09%) had a history of contraceptive use prior to their current pregnancy. One-hundred and ninety-six (48.88%) pregnant women had a history of regular use of a computer or watching television for more than 2 hours per day (Table 2).

## Prevalence of refractive error and its associated factors

The overall prevalence of RE among pregnant women in this study was 35.66% (95% CI: 30.95%-40.37%). The most frequent type of RE was myopia (64.34%) followed by hyperopia (31.47%) and antimetropia (4.20%). The prevalence of myopia, hyperopia and antimetropia in both or either eyes from the total study participants was 22.94% (92/401), 11.22% (45/401), and 1.50% (6/401), respectively. Higher prevalence of RE was found among pregnant women with GDM (78.95%), PIH (89.29%), those who regularly used a computer or watched TV (38.78%).

On bivariable logistic regression analysis, RE was associated with the age of study participants, residence, educational status, occupation, income, parity, gestational age, GDM, PIH, family history of spectacle use, regular use of computers or watching TV, medication history, history of contraceptive use, sleep duration, history of lifetime coffee drink, and hematocrit

**Table 1. Socio-demographic characteristics of pregnant women attending ANC unit at the University of Gondar Comprehensive Specialized Hospital, Northwest Ethiopia, 2020(n = 401).**

| Variables | Frequency | Percentage (%) |
|---|---|---|
| **Age in years [c]** | Median = 27(IQR = 24–31) | |
| **Religion** | | |
| Christian | 320 | 79.80 |
| Muslim | 81 | 20.20 |
| **Residence** | | |
| Urban | 335 | 83.54 |
| Rural | 66 | 16.46 |
| **Educational status** | | |
| Cannot read & write | 53 | 13.22 |
| Grade 1–8 | 88 | 21.95 |
| Grade 9–12 | 109 | 27.18 |
| College/University | 151 | 37.66 |
| **Occupation** | | |
| Government employee | 104 | 25.94 |
| Private employee | 79 | 19.70 |
| Merchant | 40 | 9.98 |
| Housewife | 135 | 33.67 |
| Others* | 43 | 10.72 |
| Monthly income [c] (ETB) | Median = 3000(IQR = 1200–45000) | |

C = continuous variable expressed in median and IQR, ETB = Ethiopian Birr, IQR = interquartile range,

*others = farmers, daily workers and unemployed.

level (p < 0.25). However, in the final model, only age, educational level, parity, gestational age and regular use of computers or watching TV were significantly associated with RE (p ≤ 0.05).

For a unit increase in age of study participants, the odds of developing RE was increased by 1.3 times (AOR = 1.31; 95% CI: 1.16–1.48). RE was increased by 15% (AOR = 1.15, 95% CI: 1.08–1.22) for each week increase of gestational age. The odds of getting RE was 6.2 times (AOR = 6.12, 95% CI: 2.46–15.59) higher among study participants who regularly used computers or watched television than the non-users. Study participants with 9–12 education level had 90% (AOR = 0.10, 95% CI: 0.02–0.54) reduced odds of refractive error than those who cannot read and write. A one-unit increase in the number of parity of study participants led to a 3.20 fold (AOR = 3.17, 95% CI: 1.92–5.25) increase in the odds of developing RE (Table 3).

## Discussion

Pregnancy is a physiological condition responsible for a variety of alterations in all body systems including the visual system. These physiological stresses during pregnancy are possibly due to changes in the hormonal, immunological, metabolic, hematologic, and cardiovascular status of the body [44,45]. Even though most of the ocular (refractive) changes are temporary (physiological) that will return to normal after delivery, there are also permanent (pathological) changes persisting after a postpartum period that may require an intervention [46]. To the best of our knowledge, there is no established study on the prevalence of RE among pregnant women in Ethiopia.

In this study, the overall prevalence of RE was 35.66% (95% CI: 30.95%-40.37%) which is lower than studies in India (65%) [11], Nigeria (90%) [10], and South India (77.50%) [12]. This variation might be accounted for the differences in study design used (cross-sectional in our

**Table 2. Clinical, substance use and pregnancy-related characteristics of pregnant women attending ANC unit at the University of Gondar Comprehensive Specialized Hospital, Northwest Ethiopia, 2020(n = 401).**

| Variables | Frequency | Percentage (%) |
|---|---|---|
| **Parity** [c] | 1(0–2) | |
| **Gestational age** [c] **(in weeks)** | 30(24–36) | |
| **Gestational diabetes mellitus** | | |
| Yes | 19 | 4.74 |
| No | 382 | 95.26 |
| **Pregnancy-induced hypertension** | | |
| Yes | 28 | 6.98 |
| No | 373 | 93.02 |
| **Family history of spectacle use** | | |
| Yes | 36 | 8.98 |
| No | 365 | 91.02 |
| **Regular use of computer or television** | | |
| Yes | 196 | 48.88 |
| No | 205 | 51.12 |
| **Medication history** | | |
| Yes | 38 | 9.48 |
| No | 363 | 91.52 |
| **Contraceptive use** | | |
| Yes | 253 | 63.09 |
| No | 148 | 36.91 |
| **Sleep duration** | | |
| Short | 47 | 11.72 |
| Optimal | 252 | 62.84 |
| Long | 102 | 25.44 |
| **Ever drink alcohol** | | |
| Yes | 270 | 67.33 |
| No | 131 | 32.67 |
| **Currently drink alcohol** | | |
| Yes | 176 | 65.19 |
| No | 94 | 34.81 |
| **Ever drink coffee** | | |
| Yes | 339 | 84.54 |
| No | 62 | 15.46 |
| **Currently drink coffee** | | |
| Yes | 276 | 81.42 |
| No | 63 | 18.58 |
| **Perceived stress level** | | |
| Low | 75 | 18.70 |
| Moderate | 294 | 73.32 |
| High | 32 | 7.98 |
| Hematocrit level [c] (%) | 39(37–41) | |

C: Continuous variable.

study) and socioeconomic factors. For instance, access to use computers or other digital devices that can precipitate the occurrence of RE in Ethiopia is very low as compared to other studies in developed countries. Therefore, these factors might be responsible for the observed discrepancy.

**Table 3. Bivariable and multivariable logistic regression analysis of factors associated with refractive error among study participants attending at ANC unit of University of Gondar Comprehensive Specialized Hospital, Northwest Ethiopia, 2020 (n = 401).**

| Variables | Refractive error | | OR (95%CI) | |
|---|---|---|---|---|
| | Yes N (%) | No N (%) | COR | AOR |
| **Age (years)** | 31(28–35) | 25(23–28) | 1.39(1.29–1.48) | 1.31(1.16–1.48)* |
| **Residence** | | | | |
| Urban | 108(32.24) | 227(67.76) | 0.42(0.25–0.72) | 0.76(0.21–2.77) |
| Rural | 35(53.03) | 31(46.97) | 1.00 | 1.00 |
| **Educational status** | | | | |
| Can't read & write | 32(60.38) | 21(39.62) | 1.00 | 1.00 |
| Grade 1–8 | 31(35.23) | 57(64.77) | 0.36(0.18–0.72) | 0.63(0.17–2.33) |
| Grade 9–12 | 11(10.09) | 98(89.91) | 0.07(0.03–0.17) | 0.10(0.02–0.54)* |
| College/University | 69(45.70) | 82(54.30) | 0.55(0.29–1.04) | 0.95(0.19–4.76) |
| **Occupation** | | | | |
| Gov't employee | 44(42.31) | 60(57.69) | 1.62(0.95–2.77) | 0.88(0.22–3.51) |
| Private employee | 31(39.24) | 48(60.76) | 1.43(0.80–2.56) | 1.44(0.39–5.28) |
| Merchant | 12(30.00) | 28(70.00) | 0.95(0.44–2.05) | 0.43(0.09–1.95) |
| House wife | 42(31.11) | 93(68.89) | 1.00 | 1.00 |
| Others** | 14(32.56) | 29(67.44) | 1.07(0.51–2.23) | 1.06(0.20–5.48) |
| **Income (ETB)** | 4000(2000–5000) | 3000(500–4200) | 1.00(1.00–1.03) | 1.00(0.98–1.00) |
| **Parity** | 2(2–3) | 0(0–1) | 4.57(3.39–6.16) | 3.17(1.92–5.25)* |
| **Gestational age (week)** | 34(28–36) | 28(20–32) | 1.12(1.08–1.16) | 1.15(1.08–1.22)* |
| **Gestational DM** | | | | |
| Yes | 15(78.95) | 4(21.05) | 7.44(2.42–22.88) | 0.63(0.10–3.92) |
| No | 128(33.51) | 254(66.49) | 1.00 | 1.00 |
| **PIH** | | | | |
| Yes | 25(89.29) | 3(10.71) | 18.01(5.33–60.83) | 2.03(0.39–10.56) |
| No | 118(31.64) | 255(68.36) | 1.00 | 1.00 |
| **Family history of spectacle use** | | | | |
| Yes | 21(58.33) | 15(41.67) | 2.79(1.39–5.60) | 2.30(0.72–7.40) |
| No | 122(33.42) | 243(66.58) | 1.00 | 1.00 |
| **Regular use of a computer or TV** | | | | |
| Yes | 76(38.78) | 120(61.22) | 1.30(0.87–1.97) | 6.19(2.46–15.59)* |
| No | 67(32.68) | 138(67.32) | 1.00 | 1.00 |
| **Medication history** | | | | |
| Yes | 26(68.42) | 12(31.58) | 4.56(2.22–9.35) | 1.91(0.50–7.31) |
| No | 117(32.23) | 246(67.77) | 1.00 | 1.00 |
| **Contraceptive use** | | | | |
| Yes | 131(51.78) | 122(48.22) | 12.17(6.42–23.07) | 2.317(0.817–6.60) |
| No | 12(8.11) | 136(91.89) | 1.00 | 1.00 |
| **Sleep duration** | | | | |
| Short | 26(55.32) | 21(44.68) | 6.19(2.85–13.45) | 1.932(0.49–7.65) |
| Optimal | 100(39.68) | 152(60.32) | 3.29(1.84–5.87) | 0.62(0.25–1.55) |
| Long | 17(16.67) | 85(83.33) | 1.00 | 1.00 |
| **Ever drink coffee** | | | | |
| Yes | 126(37.17) | 213(62.83) | 1.57(0.86–2.85) | 2.16(0.73–6.35) |
| No | 17(27.42) | 45(72.58) | 1.00 | 1.00 |
| **Hematocrit level [c] (%)** | 39.4(37.4–41) | 38.85(37–40.8) | 1.04(0.98–1.11) | 0.97(0.87–1.09) |

AOR = Adjusted odds ratio, CI = confidence interval, COR = crude odds ratio, ETB = Ethiopian birr, Gov't = government, TV = Television, * = p-value $\leq$ 0.05,

**others = farmers, daily workers and unemployed.

The prevalence of RE in our study is higher than other studies in Saudi Arabia (25.36%) [8], India (10.50%) [47], Ghana (21.60%) [48], and Rwanda (18.90%) [49]. This discrepancy might be because of the difference in the study population, in which our study population were pregnant women but the general population for other studies. Therefore, the observed higher prevalence of RE in this study might be due to pregnancy-induced endocrine changes. An increased level of estrogen and progesterone during pregnancy is associated with a higher chance of developing refractive errors (either myopia or hyperopia). The reason for this incidence of RE is because of fluid retention in the cornea triggered by estrogen and progesterone. Consequently, fluid retention in the corneal region leads to corneal edema, progressive loss of corneal sensitivity, increased central corneal thickness and curvature, increased thickness of the lens and transient loss of accommodation, and finally end up with RE [39,44,50,51].

The association of maternal age and RE was established in our study. A unit increase in the age of the study participant is associated with a 1.3 times increment of RE. This finding is supported by previous population-based studies in the USA [18], South India [19], India [47], Sri Lanka [52], China [38,53], and Canada [54]. This relationship might be because of an increase in the firmness & hardening of the nucleus of the lens and thickness of cortex of the lens with age secondary to increased insoluble protein molecules and thus, decreases refractive index. With increasing age, the crystalline lens and corneal curvature become steeper and result in RE [18,55,56].

In our study, for each week increase in the gestational age, the odd of having RE was increased by 1.15 times. This finding is comparable with other studies in the USA [57], Iran [23], India [12], and Nigeria [10,24]. The reason for this change might be due to an increased thickness and curvature of the cornea from first to third trimesters of pregnancy following a higher rate of fluid retention triggered by hormones mainly estrogen and progesterone [50]. Aldosterone is also highly secreted during pregnancy and has a synergistic effect with estrogen to cause reabsorption of excess sodium from the renal tubules. This in turn causes the maternal blood volume to rise that leads to accumulation of fluid in the cornea [51]. Other supplemental causes for this incident may also be because of an increased in metabolic rates and high oxygen intake, production of extra red blood cells as a result of increased activities of the bone marrow due to activation of erythropoietin by Placental Lactogen and the rise of cardiac output and heart rate to go with the excess fluid volume [51,58].

A unit increase in the number of births (parity) among pregnant women was associated with increased odds of RE. The finding of our study is in agreement with the finding of a study in Croatia (28) in which, corneal edema, thickness, and curvature were more elevated in multiparous than nulli and primiparous women during pregnancy. The other plausible reason that plays a role for this event is hormonal influences on the successive pregnancies which will upset the physiological refraction system of the eyes.

The likelihood of developing RE among study participants who had a history of regular use of a computer or watching television was higher than those who didn't use. This is fairly consistent with a study in Ethiopia [35], China [59], Iran [1], India [36], and Egypt [60]. This association might be anticipated because staring on the computer, watching television and using other digital devices for a long time (more than 2 hours per day) could induce prolonged accommodation and muscle fatigue (eyestrain) that may result in a transient myopic shift [61,62]. Besides, this positive association might be coined from decreased blink rate and incomplete blinks of eyes while staring on computer or television for a prolonged time. When the eyes are exposed to an incomplete and reduced frequency of blinking during exhaustive use of digital instruments, the tear film doesn't properly run across the whole cornea. This will bring about dry eye symptoms and alter the refractive power of the cornea [62,63].

Our study found that those study participants with an educational level of grade 9–12 were less likely to experience RE than those who couldn't read and write. Contrarily, previous studies

in the United Kingdom [21], Turkey [22], China [64], and Singapore [65] showed a direct association between RE and advanced level of education. The inverse association in our study might be accounted by relatively smaller sample size, less exposure for risk factors (since most of them are housewives), and environmental and socio-economic conditions like low access to use computers or other digital devices as compared to other studies in developed countries.

The present study was cross-sectional and therefore, we are unable to show the cause-effect relationship between independent variables and refractive error. We didn't also apply cycloplegic refraction test to control the accommodation ability of the lens because of its least feasibility in our institution even though its effect is very little. Lastly, we did not collect data regarding sunlight exposure level and time spent in the outdoor which might affect the outcome variable.

## Conclusion

The prevalence of RE was high among pregnant women, of which myopia was the most common variety. Advanced age, increased gestational age, increased parity, and regular use of a computer or watching television were factors significantly and positively associated with RE among pregnant women. Apart from providing other maternal health services, a routine screening, and evaluation of pregnant women for RE during ANC visit is also recommended.

## Supporting information

**S1 File.**
(DOCX)

## Acknowledgments

We would like to acknowledge the study participants, data collectors and supervisors for their willingness, valuable support and assistance during this work. We are deeply thankful to the staff members of the Department of Optometry at the UoGCSH for their provision of different data collection tools and taking part in the data collection.

## Author Contributions

**Conceptualization:** Mengistie Diress.

**Data curation:** Mengistie Diress, Yigizie Yeshaw, Minychil Bantihun, Baye Dagnew, Adugnaw Ambelu, Mohammed Abdu Seid, Yonas Akalu.

**Formal analysis:** Mengistie Diress.

**Funding acquisition:** Mengistie Diress.

**Investigation:** Mengistie Diress, Yigizie Yeshaw, Minychil Bantihun, Baye Dagnew, Adugnaw Ambelu, Mohammed Abdu Seid, Yonas Akalu.

**Methodology:** Mengistie Diress, Yigizie Yeshaw, Minychil Bantihun, Baye Dagnew, Adugnaw Ambelu, Mohammed Abdu Seid, Yonas Akalu.

**Project administration:** Mengistie Diress.

**Resources:** Mengistie Diress.

**Supervision:** Mengistie Diress, Baye Dagnew, Adugnaw Ambelu, Yonas Akalu.

**Visualization:** Yigizie Yeshaw, Minychil Bantihun, Baye Dagnew, Adugnaw Ambelu, Mohammed Abdu Seid, Yonas Akalu.

**Writing – original draft:** Yigizie Yeshaw, Minychil Bantihun, Baye Dagnew, Adugnaw Ambelu, Mohammed Abdu Seid, Yonas Akalu.

**Writing – review & editing:** Yigizie Yeshaw, Minychil Bantihun, Baye Dagnew, Adugnaw Ambelu, Mohammed Abdu Seid, Yonas Akalu.

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
