## [Decision Letter · Decision Letter 0]

22 Oct 2020

PONE-D-20-30136

Refractive error and its associated factors among pregnant women attending antenatal care unit at University of Gondar Comprehensive Specialized Hospital, Northwest Ethiopia

PLOS ONE

Dear Dr. Diress,

Thank you for submitting your manuscript to PLOS ONE. After careful consideration, we feel that it has merit but does not fully meet PLOS ONE’s publication criteria as it currently stands. Therefore, we invite you to submit a revised version of the manuscript that addresses the points raised during the review process.

Two experts in the field handled your manuscript, and we are very thankful for their time and efforts. Although interest was found in your study, some important comments arose that must be addressed. Notably, the methods need to be clearly described; there is additional data that could improve interest in this study; and the discussion needs to focus the current study in light of the existing literature. It is highly recommended that the manuscript be proofed by a professional editing firm before resubmission.

We look forward to receiving your revised manuscript.

Kind regards,

Frank T. Spradley

Academic Editor

PLOS ONE

2. In your Methods section, please provide additional information about the participant recruitment method and the demographic details of your participants. Please ensure you have provided sufficient details to replicate the analyses such as:   

-    a description of any inclusion/exclusion criteria that were applied to participant recruitment,

-    a table of relevant demographic details,

-    a statement as to whether your sample can be considered representative of a larger population,

-    a description of how participants were recruited, and

-       descriptions of where participants were recruited and where the research took place.

Reviewers' comments:

Reviewer's Responses to Questions

**Comments to the Author**

1. Is the manuscript technically sound, and do the data support the conclusions?

Reviewer #1: Yes

Reviewer #2: Partly

2. Has the statistical analysis been performed appropriately and rigorously? 

Reviewer #1: Yes

Reviewer #2: Yes

3. Have the authors made all data underlying the findings in their manuscript fully available?

Reviewer #1: Yes

Reviewer #2: No

4. Is the manuscript presented in an intelligible fashion and written in standard English?

Reviewer #1: No

Reviewer #2: No

5. Review Comments to the Author

Reviewer #1: Important issue taken up. But it needs improvement. See the attached folder for specific comments and address them accordingly.

Final appraisal:

1. Improve writing. You may seek help from professional editing.

2. Rural-urban difference in prevalence of myopia in your study would be interesting.

3. Look for any association of time spent outdoors or sun exposure and refractive error prevalence.

Reviewer #2: While the study is interesting, the the paper needs improvements throughout all sections. However, the methodology and discussion sections are the ones that need major improvement, as the procedures are not adequately explained and the message is clear. The methodology sections need to be reformulated to emphasize the procedure of clinical data (refraction) and other data collections. The methods/procedure of data collection is not described in sufficient detail. Several references are not properly cited and/or are outdated. It needs to be replaced and re-arranged.

Detail comments attached

6. PLOS authors have the option to publish the peer review history of their article (what does this mean?). If published, this will include your full peer review and any attached files.

Reviewer #1: **Yes: **Bhim RAI

Reviewer #2: **Yes: **Indra P Sharma

---

## [Author Response · Author response to Decision Letter 0]

18 Nov 2020

Point by point response for editor/reviewers comments

Manuscript title: Refractive error and its associated factors among pregnant women attending antenatal care unit at University of Gondar Comprehensive Specialized Hospital, Northwest Ethiopia. 

Manuscript ID: PONE-D-20-30136

Dear editor/reviewers: Thank you for giving us the chance to revise the manuscript. We have addressed all the concerns raised and these modifications are also incorporated in the revised manuscript.

A. Journal requirements:

Author’s response: We have prepared the manuscript based on PLOS ONE's style requirements.

2. In your Methods section, please provide additional information about the participant recruitment method and the demographic details of your participants. Please ensure you have provided sufficient details to replicate the analyses such as:

• A description of any inclusion/exclusion criteria that were applied to participant recruitment,

• A table of relevant demographic details,

• A statement as to whether your sample can be considered representative of a larger population,

• A description of how participants were recruited, and

• Descriptions of where participants were recruited and where the research took place.

Author’s response: All information how and where our participants were recruited, demographic details of the participants and the inclusion/exclusion criteria were included in the revised manuscript. Our sample was sufficient to draw conclusions for our source population. We have calculated the sample size based on single population proportion formula by using a proportion (P) of 0.5 to get the largest possible minimum adequate sample size since there were no previous studies in Ethiopia. We have also used random sampling technique to recruit each study participant. Therefore, our sample was representative of the target population (pregnant women who visited ANC service of the University of Gondar Compressive Specialized Hospital).

3. PLOS requires an ORCID ID for the corresponding author in Editorial Manager on papers submitted after December 6th, 2016. Please ensure that you have an ORCID ID and that it is validated in Editorial Manager. To do this, go to ‘Update my Information’ (in the upper left-hand corner of the main menu), and click on the Fetch/Validate link next to the ORCID field. This will take you to the ORCID site and allow you to create a new ID or authenticate a pre-existing ID in Editorial Manager. Please see the following video for instructions on linking an ORCID ID to your Editorial Manager account: https://www.youtube.com/watch?v=_xcclfuvtxQ

Author’s response: ORCID ID for the corresponding author was created which is 0000-0003-4186-4991, and the link to public record is https://orcid.org/0000-0003-4186-4991. 

Author’s response: The ethics statement was moved to the methods section.

B. Response to reviewers comments

For reviewer # 1

1. Is the manuscript presented in an intelligible fashion and written in Standard English? You may seek help from professional editing

Author’s response: Thank you. We have made extensive English editing throughout the document. 

2. Page (P) 2, Line (L) 27: Instead of writing “significantly significant factors” just write significant factors.

Author’s response: Thank you. We have deleted the word “significantly” as suggested (page (P) 1, Line (L)-28).

3. Did you consider finding out the rural-urban difference in myopia prevalence among the participants? Rural-urban difference in myopia prevalence in your study and compared with findings in other studies conducted in similar developing countries would be interesting to look at. 

Author’s response: Thank you for raising this important point. We have checked the association between residence and our outcome variable (RE) in bivariable and multivariable logistic regression analysis. However, the adjusted odds ratio of the variable residence was 0.76 with 95% CI of 0.21-2.77 in our final analysis model. Since the 95% CI includes 1, the association of residence to RE is non-significant (p>0.05) and showed that there is no as such significant difference between urban and rural residents in regarding to RE. Hence, considering residence as stratifying variable will not add new information. 

4. Did you consider looking at the time spent outdoors as causative factors in your study? Your approach seems to have been mainly based on the compensatory mechanism of developing myopia. Currently there is emphasis on the Dopamine theory of myopia development. On this line have you looked if exposure to sunlight or time spent outdoors were related to development of refractive errors?

Author’s response: Thank you. We didn’t include the time spent outdoors as causative factors directly in our study. However, we have tried to assess its impact based on the occupation of participants, which is highly related to their time spent outdoors. Unfortunately, the variable occupation was not statistically significant (P>0.05) (see table 3) in our final analysis model. We have also put it as limitation of the study (P13, L315-316).

5. Considering the myopia prevalence in some countries ranging up to higher 80%, even up to 96.5% in some population group, the refractive error prevalence of 35.66% is not high.

Author’s response: Thank you. Even though the overall prevalence of refractive error in our study is lower than the previous studies mentioned, still it is higher than what is intended to be in 2020 (2.9%). Besides, the prevalence of myopia in our study was 64.34% (P9, L202) which is fairly similar with the numbers you mentioned. Therefore, the prevalence of refractive error and myopia in our study were high. 

6. Is cataract the commonest cause of blindness and visual impairment in Ethiopia, second being refractive error?

Author’s response: Thank you very much. Cataract is the leading cause of both blindness (49.9%) and visual impairment (42.3%). Refractive error is the second cause for visual impairment (33.4%) but the third cause of blindness (7.8%) in Ethiopia. We have also stated this sentence in the revised manuscript (P3, L50-52).

7. P3, L68: give space after hyperopia.

Author’s response: Thank you. We have modified it based on your suggestion (P3, L59).

8. P4, L70: You need not define disabilities in the bracket. It is understood. Other examples are L46-48, where you have defined the terms in bracket. These are not required. All these redundant or loose writings are grading down your article.

Author’s response: Thank you. We have removed all of the terms in bracket in the revised manuscript. 

9. P4, L69-73: Modify this sentence. It is too lengthy, and some link or words are missing before the retinal detachment.

Author’s response: Thank you very much. We have modified the sentence in the revised manuscript as suggested (P3, L61-65).

10. P5, L98: It is not clear what you want to say here.

Author’s response: Thank you. It is to mean that our outcome variable was refractive error in which the participants were considered to have refractive error, if they were either myopic (the spherical equivalent ≤ –0.50 dioptres) or hyperopic (the spherical equivalent ≥ +0.50D) or both problems in either or both eyes. If the spherical equivalent (SE) is in between -0.50D and +0.50D in either or both eyes, they are considered as normal. We have also included this modification in the revised manuscript (P6, L106-110).

11. P5, L99: It is tricky here. Do you mean to say those women more myope than -0.50D, and hyperopia more than +0.50 D were included? You need to be clearer in this sentence.

Author’s response: Thank you. We have modified the statement to make it more clearly to readers (P6, L106-110).

12. P6, L126: What is khat?

Author’s response: Thank you. Khat (Catha edulis) is a flowering plant native to Ethiopia which contains the alkaloid cathinone, a neuro-stimulant substance. For clarification, we stated as “Catha edulis” as an alternative name to Khat in the revised manuscript (P6, L113).

13. P7, L144-147: The data collectors were trained so why was it necessary to translate the questionnaire from English to local language and back to English?

Author’s response: Thank you. This was done to check the meaning consistency of the questioner. The mother tongue for data collectors and the participants is Amharic language. Therefore, to create common understanding and to avoid any possible misinterpretation during the interview, we prefer to translate it into Amharic language.

14. P10, L192-194: This sentence does not make any sense or even contrasting your own findings. Please explain.

Author’s response: Thank you. it is to mean that study participants with grade 9-12 education level had 0.1 times reduced odds of RE compared to the reference group. As we stated in the manuscript, the above interpretation can also be expressed as study participants with 9-12 education level had 90% (AOR=0.10, 95%CI: 0.02-0.54) reduced odds of refractive error than those who cannot read and write (P13, 18-19, L220-222, 297-304). Therefore, our interpretation is in line with our finding. 

15. P11, L216: What is RE? You have not defined it earlier. You have used refractive errors and RE interchangeably without defining it. If you wish to use abbreviations define it in the first mention and be consistent with it.

Author’s response: Thank you very much. We have used RE just to say refractive error and now we have corrected it as you suggested in the revised document. 

16. P11, L235: Provide space after progesterone.

Author’s response: Thank you. We have corrected based on your suggestion (P17, L272).

17. Are there any limitations in your study that you would like to mention at the end of the Discussion?

Author’s response: Thank you. As we stated in the last paragraph of discussion section, the present study has the following limitations (P19, L305-311). First, the study was cross-sectional study and hence we are unable to show cause-effect relationship between independent variables and refractive error. Second, we didn’t measurement the axial length of the eyes of study participants that may affect our outcome variable due to resource scarcity. Third, we didn’t apply cycloplegic refraction test to regulate accommodation ability of the lens because of its least feasibility in our institution even though its effect is very little. Lastly, we did not collect data regarding sunlight exposure level and time spent in the outdoor which might have an effect on the outcome variable.

For Reviewer #2

1. Is the manuscript technically sound, and do the data support the conclusions?

Author’s response: Thank you. We have tried to make the manuscript more technically sound and our data supports our conclusion in the revised manuscript.

2. Is the manuscript presented in an intelligible fashion and written in Standard English?

Author’s response: Thank you. We have made extensive English editing throughout the document. 

3. The paper needs improvements throughout all sections. However, the methodology and discussion sections are the ones that need major improvement, as the procedures are not adequately explained and the message is clear. The methodology sections need to be reformulated to emphasize the procedure of clinical data (refraction) as well as visual impairment. The methods/procedure of data collection is described in sufficient detail. Several references are not properly cited and/or are outdated. It needs to be replaced and re-arranged.

Author’s response: Thank you very much. As we can see in the track changes of the document, we tried to address all the comments and included these corrections in the revised manuscript.

4. It would be appropriate to give an global estimate of refractive error(RE), prevalance of RE on pregnancy, imapct, known associations and establish gap.

Author’s response: Thank you. We have reorganized it based on your suggestion (P3-4, L45-77).

5. Lines 44-48: Sentences not necessary . Please delete.

Author’s response: Thank you. We have deleted the sentence in the revised manuscript. 

6. Lines 49-51: Estimates are outdated and reference [5-6] are old. Please use latest estimates (Flaxman SR, 2017, Lancet)

Author’s response: Thank you. We have corrected the reference as suggested (P3, L46).

7. Line 53: Reference 3 is old. Please add recent estimates.

Author’s response: Thank you. We had modified the references in the revised manuscript (P3, L49)

8. Lines 51-54: The sentence is confusing; as it talks of prevalence in Ethopia and refers to study in Timor-Leste.

Author’s response: Thank you very much. Sorry for the mistake we made. The relevant reference was cited and we have also rewrite the sentence to make it clear in revised manuscript (P3, L50-52).

9. Line 55: Please use updated reference and add available reference on prevalence on refractive error in pregancy.

Author’s response: Thank you. We had made corrections regarding your suggestion in the revised manuscript (P3, L53-60). 

10. Lines 56-64: This section should go to discussion.

Author’s response: Thank you. Modification was mad based on your suggestion and the paragraph was incorporated in the discussion section.

11. Lines 78-80: The hospital based prevalence may not be adequate to generalise to general population in Ethopia.

Author’s response: Thank you. We accepted your comment and the word “country” was replaced by “study setting” (P4, L77).

12. Lines 86-87: Why was pegnant women with medical conditions and previous RE excluded? Was women with pre-existing RE excluded from the study. Excluding women with eye problem/RE before will underestimate prevalence of RE in pregant women.

Author’s response: Thank you. We excluded the pregnant women with the above medical conditions before pregnancy (diabetes and hypertension before being pregnant) because these conditions are known to increases the chance of having refractive error. We also excluded pregnant women who had RE before pregnancy. This is because our aim was determining the prevalence of RE during pregnancy. Actually the numbers of pregnant women who had RE before pregnancy and excluded in this study were only 3. Therefore, excluding these women may not significantly underestimate the outcome variable. 

13. Lines 99-100: Please cite a reference why you chose to consider SE of 0.5 for myopia and hypermetropia.

Author’s response: Thank you. Relevant references were cited as suggested (P6, L110). 

14. Lines 106-131: The operational definition may come as a supplementary file. Extensive definition is not necessary. As degrees of RE was not assessed in the study nor discussed in this paper, it need not be mentioned.

Author’s response: Thank you very much. We have tried to minimize and merge the operational definitions as suggested in the revised manuscript (P7, L116-135). 

15. 132-137: The ‘Data Collection Procedure’ is not adequately described. Please attach the questionnaire and proforma for data collection as as supplement files. Was the questionaire validiated? Since refractive error is the primary outcome measure, please explain in detail how refraction was performed (cycloplegic/non-cycloplegic).

Author’s response: Thank you. We used a validated questioner which was prepared from different literatures after extensive searching. We performed a non-cycloplegic refraction because of its least feasibility in our setup. Cycloplegic refraction was necessarily indicated for patients suspected for pseudomyopia, accommodative esotropia and for those aged less than 20 years. In our study, most of the participants were aged more than 22 years and hence the effect of accommodation for outcome variable is less significant. There was also no indications to perform cycloplegic test in our study participants. We have included all the concerns raised in the revised manuscript (P7-8, L136-148).

16. Please use decimals uniformly throughout the result section.

Author’s response: Thank you. As you can see in the track change document, we have made the decimals uniform throughout the revised manuscript.

17. Lines 170-175: The paragraph is the repatition of Fig 1 and does provide same results. Please delete either one of it. 

Author’s response: Thank you. We have deleted the figure as suggested.

18. Lines 210-211: If the cut-off pont differs, comparion of RE is not possible. Please compare with those with similar cut-off points.

Author’s response: Thank you. In our discussion section, we had modified the cut-off point as a means of discrepancy in the revised manuscript (P16, L245-248).

19. Lines 212-215: Comparision of RE with normal population of the study site is important. Comparison of pregnant women (only female) from the study site to general population (male/female) from other country is not appropriate.

Author’s response: Thank you. Since there are limited previous studies on the prevalence of refractive error among pregnant women, we are forced to discuss our finding with the finding non pregnant women and the general population. In the absence of sufficient studies in the area, we hope, comparing our finding with the above population will be important to appreciate the difference and take intervention accordingly. If you are not convinced with our argument, it is possible to delete that part.

---

## [Decision Letter · Decision Letter 1]

17 Dec 2020

PONE-D-20-30136R1

Refractive error and its associated factors among pregnant women attending antenatal care unit at the University of Gondar Comprehensive Specialized Hospital, Northwest Ethiopia

PLOS ONE

Dear Dr. Diress,

Thank you for submitting your manuscript to PLOS ONE. After careful consideration, we feel that it has merit but does not fully meet PLOS ONE’s publication criteria as it currently stands. Therefore, we invite you to submit a revised version of the manuscript that addresses the points raised during the review process.

There are some remaining comments from the reviewers that require your attention.

We look forward to receiving your revised manuscript.

Kind regards,

Frank T. Spradley

Academic Editor

PLOS ONE

Reviewers' comments:

Reviewer's Responses to Questions

**Comments to the Author**

1. If the authors have adequately addressed your comments raised in a previous round of review and you feel that this manuscript is now acceptable for publication, you may indicate that here to bypass the “Comments to the Author” section, enter your conflict of interest statement in the “Confidential to Editor” section, and submit your "Accept" recommendation.

Reviewer #1: All comments have been addressed

Reviewer #2: (No Response)

2. Is the manuscript technically sound, and do the data support the conclusions?

Reviewer #1: Yes

Reviewer #2: Yes

3. Has the statistical analysis been performed appropriately and rigorously? 

Reviewer #1: Yes

Reviewer #2: Yes

4. Have the authors made all data underlying the findings in their manuscript fully available?

Reviewer #1: Yes

Reviewer #2: Yes

5. Is the manuscript presented in an intelligible fashion and written in standard English?

Reviewer #1: Yes

Reviewer #2: No

6. Review Comments to the Author

Reviewer #1: The authors have responded all the points raised to my satisfaction.

Just a small point: in the revised manuscript under Limitations of the study section, the second limitation you have mentioned is that you did not measure the axial length which might have affected the outcome variables. I do not think this would affect at all because you are not defining the refractive error by axial length in your study.

Reviewer #2: Comments

The manuscript is significantly improved; however, several areas can still be improved.

Abstract:

Lines 16-17: ‘such as retinal detachment, difficulty of performing daily living activities and economic crisis’ this can be removed or rephrased.

Line 18: remove the word ‘Therefore’.

Line 30: ‘27 years with inter-quartile range of 24 to 31 years’ can be written as ’27 (IQR 24-31) years.

Manuscript

1. There are still a lot of grammatical errors. Please correct.

2. Use the abbreviation of ‘University of Gondar Comprehensive Specialized Hospital’ throughout the manuscript after its first mention in line 75-76.

3. Line 46-48: Statement not appropriate to this study. Can delete.

4. Line 141-142: Delete the line as its explained in lines 142-145.

5. Line 151: Pre-test = Pilot?

6. Line 175: Can you replace the word ‘linked’ with more appropriate word.

7. Line 179-180: Rephrase the sentences and correct the grammar.

8. Lines 240-242: Delete the lines as its already stated in introduction.

9. In the discussion, use odds ratios to describe significance rather than %.

10. Conclusion could be the past paragraph of the discussion.

Reference:

Reference 4: Please correct citation style.

7. PLOS authors have the option to publish the peer review history of their article (what does this mean?). If published, this will include your full peer review and any attached files.

Reviewer #1: No

Reviewer #2: **Yes: **Indra Prasad Sharma

---

## [Author Response · Author response to Decision Letter 1]

22 Dec 2020

Point by point response for reviewers’ comments Date: Dec 21, 2020

Manuscript title: Refractive error and its associated factors among pregnant women attending antenatal care unit at University of Gondar Comprehensive Specialized Hospital, Northwest Ethiopia. 

Manuscript ID: PONE-D-20-30136R1

Dear editor/reviewers: Thank you for giving us the chance to revise the manuscript. We have addressed all the concerns raised and these modifications are also incorporated in the revised manuscript.

Response to reviewers’ comments

For reviewer # 1

1. Just a small point: in the revised manuscript under Limitations of the study section, the second limitation you have mentioned is that you did not measure the axial length which might have affected the outcome variables. I do not think this would affect at all because you are not defining the refractive error by axial length in your study.

Author’s response: Thank you. We have agreed with your comment and removed it from the revised manuscript. 

For reviewer # 2

1. Is the manuscript presented in an intelligible fashion and written in standard English?

Author’s response: Thank you. We have made an extensive English editing throughout the revised document to improve the quality of our paper.

2. Lines 16-17: ‘such as retinal detachment, difficulty of performing daily living activities and economic crisis’ this can be removed or rephrased.

Author’s response: Thank you. We have removed the lists in the revised manuscript.

3. Line 18: remove the word ‘Therefore’.

Author’s response: Thank you. We have removed it as suggested.

4. Line 30: ‘27 years with inter-quartile range of 24 to 31 years’ can be written as ’27 (IQR 24-31) years. 

Author’s response: Thank you. We have rewritten it based on your suggestion.

5. There are still a lot of grammatical errors. Please correct.

Author’s response: Thank you. We have tried to correct grammatical errors throughout the revised manuscript.

6. Use the abbreviation of ‘University of Gondar Comprehensive Specialized Hospital’ throughout the manuscript after its first mention in line 75-76.

Author’s response: Thank you. We have corrected as suggested in the revised manuscript.

7. Line 46-48: Statement not appropriate to this study. Can delete.

Author’s response: Thank you. We have removed the unnecessary statement as suggested in the revised manuscript. 

8. Line 141-142: Delete the line as its explained in lines 142-145.

Author’s response: Thank you. We have deleted it in the revised manuscript as suggested.

9. Line 151: Pre-test = Pilot?

Author’s response: Thank you. We had conducted a pre-test not a pilot study, to assess the reliability and validity of the survey instruments/questioners prior to the actual data collection. 

10. Line 175: Can you replace the word ‘linked’ with more appropriate word.

Author’s response: Thank you. We have replaced the word ‘linked’ with “referred” in the revised manuscript (Line 170). 

11. Line 179-180: Rephrase the sentences and correct the grammar.

Author’s response: Thank you. We have rephrased and corrected the grammar as suggested. 

12. Lines 240-242: Delete the lines as its already stated in introduction.

Author’s response: Thank you. We have removed the statement in the revised manuscript as suggested.

13. In the discussion, use odds ratios to describe significance rather than %.

Author’s response: Thank you. We have modified our description of association in the discussion section of the revised manuscript.

14. Conclusion could be the past paragraph of the discussion.

Author’s response: Thank you. We have placed the limitations of the study at the last paragraph of the discussion rather than conclusion. This is because the guideline of PLOS ONE recommends the conclusion to be placed separately just next to the discussion section.

15. Reference 4: Please correct citation style.

Author’s response: Thank you. The citation style we used for all references was Vancouver, including reference 4. Reference 4 is a guideline with no volume and issue numbers. Since this reference did not add extra information beyond references 2 and 3, we have removed it in the revised manuscript

---

## [Decision Letter · Decision Letter 2]

7 Jan 2021

PONE-D-20-30136R2

Refractive error and its associated factors among pregnant women attending antenatal care unit at the University of Gondar Comprehensive Specialized Hospital, Northwest Ethiopia

PLOS ONE

Dear Dr. Diress,

Thank you for submitting your manuscript to PLOS ONE. After careful consideration, we feel that it has merit but does not fully meet PLOS ONE’s publication criteria as it currently stands. Therefore, we invite you to submit a revised version of the manuscript that addresses the points raised during the review process.

There are still a few issues with grammar and spelling errors. Please contact a copyeditor to proof this manuscript before resubmission. 

We look forward to receiving your revised manuscript.

Kind regards,

Frank T. Spradley

Academic Editor

PLOS ONE

Reviewers' comments:

Reviewer's Responses to Questions

**Comments to the Author**

1. If the authors have adequately addressed your comments raised in a previous round of review and you feel that this manuscript is now acceptable for publication, you may indicate that here to bypass the “Comments to the Author” section, enter your conflict of interest statement in the “Confidential to Editor” section, and submit your "Accept" recommendation.

Reviewer #1: All comments have been addressed

Reviewer #2: All comments have been addressed

2. Is the manuscript technically sound, and do the data support the conclusions?

Reviewer #1: Yes

Reviewer #2: Yes

3. Has the statistical analysis been performed appropriately and rigorously? 

Reviewer #1: Yes

Reviewer #2: Yes

4. Have the authors made all data underlying the findings in their manuscript fully available?

Reviewer #1: Yes

Reviewer #2: Yes

5. Is the manuscript presented in an intelligible fashion and written in standard English?

Reviewer #1: Yes

Reviewer #2: No

6. Review Comments to the Author

Reviewer #1: Thank you for addressing all my comments raised during the previous reviews. I do not have further comments. Congratulations!

Reviewer #2: All comments have been addressed. The manuscript can be accepted for publication.

However, minor grammatical errors can still be corrected.

7. PLOS authors have the option to publish the peer review history of their article (what does this mean?). If published, this will include your full peer review and any attached files.

Reviewer #1: No

Reviewer #2: **Yes: **Indra P Sharma

---

## [Author Response · Author response to Decision Letter 2]

13 Jan 2021

Response for reviewers’ comment Date: Jan 12, 2021

Manuscript title: Refractive error and its associated factors among pregnant women attending antenatal care unit at the University of Gondar Comprehensive Specialized Hospital, Northwest Ethiopia. 

Manuscript ID: PONE-D-20-30136R2

Dear editor/reviewers: Thank you for giving us the chance to revise the manuscript. We have addressed all the concerns raised and these modifications are also incorporated in the revised manuscript.

Response to reviewers’ comment

For reviewer # 2

1. Is the manuscript presented in an intelligible fashion and written in standard English?

 All comments have been addressed. The manuscript can be accepted for publication.

However, minor grammatical errors can still be corrected.

Author’s response: Thank you. We have made an English editing throughout the document to improve grammatical errors as suggested.

---

## [Editor Report · Decision Letter 3]

15 Jan 2021

Refractive error and its associated factors among pregnant women attending antenatal care unit at the University of Gondar Comprehensive Specialized Hospital, Northwest Ethiopia

PONE-D-20-30136R3

Dear Dr. Diress,

We’re pleased to inform you that your manuscript has been judged scientifically suitable for publication and will be formally accepted for publication once it meets all outstanding technical requirements.

Kind regards,

Frank T. Spradley

Academic Editor

PLOS ONE

---

## [Editor Report · Acceptance letter]

3 Feb 2021

PONE-D-20-30136R3 

Refractive error and its associated factors among pregnant women attending antenatal care unit at the University of Gondar Comprehensive Specialized Hospital, Northwest Ethiopia 

Dear Dr. Diress:

I'm pleased to inform you that your manuscript has been deemed suitable for publication in PLOS ONE. Congratulations! Your manuscript is now with our production department. 

Kind regards, 

on behalf of

Dr. Frank T. Spradley 

Academic Editor

PLOS ONE